# Tool Wear and Surface Roughness in Turning of Metal Matrix Composite Built of Al_2_O_3_ Sinter Saturated by Aluminum Alloy in Vacuum Condition

**DOI:** 10.3390/ma15238375

**Published:** 2022-11-24

**Authors:** Michał Szymański, Damian Przestacki, Paweł Szymański

**Affiliations:** 1Faculty of Mechanical Engineering, Institute of Material Technology, Poznan University of Technology, ul. Piotrowo 3, 61-138 Poznan, Poland; 2Faculty of Mechanical Engineering, Institute of Mechanical Technology, Poznan University of Technology, ul. Piotrowo 3, 61-138 Poznan, Poland

**Keywords:** metal matrix composites (MMCs), turning of MMCs, tool wear, surface roughness, cutting forces

## Abstract

Metal matrix composites (MMCs) are a special class of materials carrying combined properties that belongs to alloys and metals according to market demands. Therefore, they are used in different areas of industry, and the properties of this type of material are useful in engineering applications. Machining of such composites is of great importance to finalize the fabrication process with improved part quality. However, the process implies several challenges due to the complexity of the cutting processes and random material structure. The current study aims to examine machinability characteristics. Effects of turning a metal matrix composite built of Al_2_O_3_ sinter, saturated with an EN AC-44000 AC-AlSi_11_ alloy, are presented in this paper. In the present study, a turning process of new metal matrix composites was carried out to determine the state-of-the-art material for various engineering applications. During the turning process, the cutting forces, a tool’s wear, and surface roughness were investigated. Further, the SEM (scanning electron microscope) analysis of cutting inserts was performed. The influence of MMC structure on the machining process and surface roughness was studied. The Al_2_O_3_ reinforcements were used in different graininess. Effects of conventional turning of the metal matrix composite with Al_2_O_3_ sinter of FEPA (Federation of European Producers of Abrasives) 046 and FEPA 100 grade were compared. Results analysis of these tests showed the necessity of continuing research on turning metal matrix composites built of an AlSi alloy and Al_2_O_3_ ceramic reinforcement. The study showed the properties of MMCs that influenced machinability. In this paper, the influence of feed rate’s value on surface roughness was carried out. The significant tool wear during the turning of the MMC was proved.

## 1. Introduction

The composite is an example of an engineering material that is made of a minimum of two different components [1]. This type of material combines properties of its elements, or new and better properties could characterize it. Composites are built of a matrix and a reinforcing phase [2].

Metal matrix composites (MMCs) are engineering materials that are a mixture of a matrix made from metals such as aluminum, copper, and magnesium and reinforcement materials [3]. The reinforcing phase could be built of continuous fibers or discontinuous fibers [4]. Ceramic compounds, e.g., silicon carbide (SiC) [5], aluminum oxide (Al_2_O_3_), boron carbide (B_4_C), titanium boride (TiB_2_), and other compounds, could be mainly used as a reinforcement [6]. Due to their properties such as mechanical and thermal strength, stiffness, and wear resistance, metal matrix composites are successfully applied in various branches of industry such as automotive or aerospace. Oczoś [2] stated that the composite participation of materials used in Airbus A380 is 52%. It could be evidence of the considerable potential of this type of material [2].

Nowadays, the number of applications of MMCs is increasing [7]. It could be generally observed that the machinability of MMCs depends on technological parameters, the temperature of the machining process, and the tools’ geometry [8]. Using a source of concentrated energy [9], e.g., laser-assisted machining [10], is interesting in terms of hard-to-cut materials’ treatment [10]. Studies on the A359/20SiCP composite showed that laser-assisted machining (LAM) is a promising manufacturing method for this hard-to-cut material. The researchers [11] proved that turning in LAM conditions with optimally selected values of technological parameters, such as cutting depth and the tool’s angular distance from the laser beam, gives satisfying results of machined surface roughness in comparison with conventional turning. It was stated that the development of studies on this method in the future is reasonable [12]. However, methods of manufacturing these types of composites are still economically dissatisfying [13]. As mentioned previously, using MMCs in different branches of industry is increasing [6]. MMCs are the most used materials for producing engine parts such as pistons, valves, connecting rods, brake system elements, and drive shafts [12]. For this reason, carrying out studies on new methods and possibilities of producing and machining metal matrix composites is economically reasonable [14].

As it was mentioned, one of the directions in manufacturing MMCs is LAM. The other option is a precision molding process. Using computer simulation could give satisfying results [15]. During this process, the final product is shaped from the beginning of production. It is also related to small material allowances. Promising results could be obtained by studies on a combination of both described methods of manufacturing [12]. That combination of manufacturing methods could be an alternative to conventional fabrication methods of machine parts, e.g., gears, characterized by some problems [16].

An important category of MMCs comprises composites formed with an aluminum alloy matrix reinforced with ceramics [17]. The ceramic reinforcement could be added as 3D structures or particles [18]. The reinforcement used in 3D skeleton structures could be made of disordered ceramic fibers or particles. The porous preform produced by that technology is saturated in the pressurized state with a liquid alloy—this way, the composite is made [14]. Costs of manufacturing metal matrix composites made of continuous fibers are higher than for metal matrix composites made of discontinuous fibers [15]. There are different methods of producing MMCs. The most popular are powder metallurgy and molding methods which are able to receive shapes close to the final one [1].

Composites built of an aluminum matrix are the group of MMCs efficiently applied in the automotive, aerospace, and energy industries [13]. Al-Si and Al-Cu alloys are successfully replaced by MMCs made of an aluminum matrix [19].

Properties and structure of metal matrix composites produced by infiltration under vacuum pressure are enhanced by the properties of ceramic scaffolds, which are the primary material used as its semi-finished product. An open-porous structure should characterize the scaffold produced by following the rules, making penetration by liquid metal possible [20].

Sintering with substances responsible for making pores is the most popular method of manufacturing MMCs with ceramic scaffolds. Sizes and density of porosity could be determined by the type of material used to make pores, e.g., polyethylene, cellulose, and paraffin [21].

The first stage of manufacturing MMCs with infiltration under vacuum pressure is preparing a preform produced by mixing a ceramic material with the material used to make pores. After that, the preform is put away and dried. When the preform is placed in an oven, the paraffin or other material is gasified. Then, the infiltration process takes place. The prepared mold is filled with an Al-Si alloy under pressure [20].

As a relatively new group of composites, metal matrix composites are examples of hard-to-cut materials [22]. MMCs are characterized by heterogeneous structures. Furthermore, these materials have properties such as high hardness and abrasive resistance. Thus, machining MMCs with PCD (polycrystalline diamond), carbide, and ceramic tools should be performed. The most popular tool types used in the machining of MMCs are uncoated tungsten carbide, coated tungsten carbide, and polycrystalline diamond [5].

Tool wear takes place during the machining of metal matrix composites. It is because of heat load and the moving of particles or fibers [23]. Mainly, there is abrasive wear on the cutting edge occurring [24]. It is possible that in the case of this phenomenon taking place on the cutting edge, there are effects appearing such as micro-cutting, peeling, chipping, material discounting, microcracking, buildups, changing of material structure, and phase transitions [1].

The main problems with manufacturing pieces from MMCs are their machining because of intensive tool wear [25] and poor workpiece surface properties [26]. Researchers [25] carried out studies on the machining of MMCs reinforced with particles and whiskers. The information about the machining of metal matrix composites built of Al_2_O_3_ sinter saturated by an aluminum alloy was not found in the available literature. Conventional tool materials such as HSS cannot be used to machine MMCs. The most effective results of machining MMCs can be achieved with ceramic and diamond tools [26]. It was observed that tool wear is caused by the abrasion of reinforcement particles which are characterized by excellent hardness [27]. Reinforcement particles in MMCs have an effect on the cutting edge similar to the grinding wheel. Poor surface finish is caused by that effect [28]. According to research [29], the most satisfactory effects on machining an Al/SiC composite are reached during turning with CBN (cubic boron nitride) inserts. Another study [30] showed that the lowest tool wear was observed using PCD and CVD (Chemical Vapor Deposition Diamond) inserts. Machining of an Al-SiC MMC with different types of diamond inserts proved a significant influence on the surface finish [27].

The results of the machining process of metal matrix composites depend on many factors [31]. Palanikumar et al. focused on the influence of machining technological parameters and cutting inserts’ geometrical parameters on the geometrical structure of the MMC surface [32]. The quality of a machined surface is one of the most critical aspects of the properties of machines’ parts [33]. Additives have an influence on tool wear [34]. Thus, the machined surface quality is a significant parameter that influences the application of metal matrix composites [17]. The surface roughness after turning MMCs depends on cutting speed and feed rate. Nataraj et al. [31] observed that the Ra parameter after turning an LM6/Al_2_O_3_ composite reached the highest value when the cutting speed was 175 m/min. By decreasing the cutting speed to 125 m/min, the value of Ra increased. It was a result of greater cutting friction occurring during machining with lower cutting speed and required high shear energy. Concerning the correlation between feed rate and surface roughness, these research studies showed that the Ra parameter reached lower values while the feed rate was low [31]. The exact correlation between feed rate and surface roughness in turning an A359/B_4_C/Al_2_O_3_ composite was presented by Srivastava et al. [19]. According to Kawalec et al. [4], another phenomenon could be noticed. These research studies prove that the surface roughness of an Al/SiC composite is reduced by increasing feed rate. It is assumed that this effect results from reducing tool wear with increasing the feed rate value. This report concludes that the surface roughness is improved at the highest values of cutting speed, feed rate, and depth of cut [35].

Pramanik et al. elaborated a mechanical model to predict the values of cutting forces during the machining of aluminum-based composites [8]. Other researchers developed a model to predict cutting forces while machining a SiC MMC [36]. Additionally, studies in the literature described the machining of composites manufactured by stir casting or powder metallurgy methods [37]. It could be observed that there are many studies on models for estimating cutting forces during machining processes [38]. Experimental studies on the machining of aluminum-based metal matrix composites show that the value of cutting force depends on plowing, chip formation, and fracture and displacement of particles. According to analytical models and experimental tests [8], it could be observed that increasing the feed rate and depth of the cut causes the growth of cutting forces. There are some discrepancies between various models and experimental tests, but the trend is constant. However, studies on the influence of cutting speed on cutting forces show that increasing the cutting speed value causes a decrease in cutting forces [8].

It should be noticed that different types of MMCs were machined in the past. However, there is no information in the available literature about studies on the machining of the material described in this paper because it is a new type of MMC, cast by the vacuum method. Nowadays, MMCs are fabricated by the stir casting method. In our investigation on the conventional turning of MMCs, the results of the turning process are shown. Described studies are basic research on the machinability of metal matrix composite casting by the vacuum method, which is not available in the literature. It is important to test the machinability of that group of materials because of the increasing industrial use of MMCs. It is reasonable to get to know the results of turning MMCs in conventional conditions before studies on machining this type of material in LAM conditions. These studies are mainly focused on investigating the machinability of a new type of MMC. The metallographic examination of a metal matrix composite built of Al_2_O_3_ sinter saturated with an EN AC-44000 AC-AlSi_11_ alloy was performed. The turning process of the MMC was completed. Cutting forces, tool wear, and surface roughness were measured.

## 2. Materials and Methods

### 2.1. Metal Matrix Composite

The present studies are mainly focused on a metal matrix composite built of Al_2_O_3_ sinter saturated with an EN AC-44000 AC-AlSi_11_ alloy. The metallographic examination of FEPA 100 was prepared. The microstructure analysis was performed on NIKON Eclipse MA200 (Figure 1). The presented material is an example of the MMC group of composites. It was manufactured with a pressureless infiltration method. The reinforcement phase is built of electrocorundum, and the matrix phase is built of an AlSi_11_ alloy. The grains of ceramic reinforcement have an irregular shape (Figure 1a,c). Some agglomerates of electrocorundum grains could be seen (Figure 1a). It should be noticed that porosity is visible in the microstructure of the material. The size of the fractions is measured in Figure 1b. Figure 1b,d show enlarged porosity.

### 2.2. Turning of Metal Matrix Composite

The turning process of the metal matrix composite built of Al_2_O_3_ sinter saturated with an EN AC-44000 AC-AlSi_11_ alloy was carried out on a laboratory station with a CNC lathe DMG CTX 310 ECOLINE produced by DMG Mori Seiki and piezoelectric dynamometer MW 2006-2. The lathe is equipped with the Siemens 840D Control System. In Figure 2, the lathe with measuring equipment and the sample after the turning process is presented. The shafts were located on the long pin between the spindle and tailstock, due to the geometry of the lathe. The mandrel was clamped in the jaws and supported by a tailstock. The spindle rotated clockwise. The variable diameter forced the spindle revolutions to change to keep the cutting speed constant. Two samples of diameter *d* = 35 mm and length *l* = 44 mm made of Al_2_O_3_ FEPA 046 and Al_2_O_3_ FEPA 100 with five separate measurement sections were prepared. The main difference between the two samples was the grain size of Al_2_O_3_ ceramic: FEPA 046—355–425 µm and FEPA 100—125–150 µm. Turning process was stopped when each measurement section was machined. Cutting time *t* ranged between 10 and 12 s. for each measurement section.

Depth of cut *a_p_* and cutting speed *v_c_* were constant technological parameters. The test was carried out with variable values of feed rate *f* = 0.05 mm/rev, *f* = 0.1 mm/rev, *f* = 0.125 mm/rev, and *f* = 0.15 mm/rev. Kennametal SNGN 120408 T01020 KYS25 ceramic cutting insert was used. Before each repetition, a new cutting tool was installed. Geometry and properties of the cutting insert are shown in Table 1.

Technological parameters of the turning process are shown in Table 2. The turning process was repeated five times. The results were analyzed with arithmetic average and dispersion.

The measurements of the tool’s wear were carried out on an optical microscope ZEISS SteREO Discovery.V20 (Figure 3, Carl Zeiss AG, Oberkochen, Germany). The cutting inserts were located in a small holder. The microscope is able to digitally measure the tool’s wear. Tests were carried out on the cutting edge.

The 3D surface roughness measurements of the workpiece were carried out with a HOMMEL-ETAMIC T-8000 profilometer (Jenoptik AG, Jena, Germany) equipped with a TKL 100/17 measuring tip (Figure 4). The shafts were located in a prism on an automatic moving worktable. With HOMMEL-ETAMIC T-8000, it is possible to create 3D measurements of surface roughness. The tests were carried out according to ISO 11562:1996. The value of *l_p_* was 4.8 mm, *λ_c_* filter—0.8 mm. The value of *λ_s_* was not defined. The mapping section was set to 1.5 mm, and the traverse length was set to 80 µm. The value of traverse speed was set at 0.05 mm/s.

## 3. Results and Discussion

### 3.1. Tool’s Wear after Machining Metal Matrix Composite

The studies were carried out with the methodology described in the previous paragraph. The research was planned according to the analyzed literature. Definition of future research courses was the main aim of these studies. Figure 5 shows the microscopic photo of the tool’s wear measurement areas of the KENNAMETAL SNGN 120408 T01020 KYS25 cutting insert. The A_γ_, A_α_, and A’_α_ are marked in these pictures.

In Figure 6a,b, the cutting insert after turning with *f* = 0.05 mm/rev of the FEPA 046 sample is shown. The abrasive wear is observed on the *A_γ_*, *A_α_*, and *A’_α_* surfaces. There is no built-up effect. The crater is observed on the cutting edge. The value of the tool’s wear is equal to *VB* = 0.77 mm. Figure 6c,d show cutting inserts after turning with the same value of feed rate *f* but on the FEPA 100 sample. In that example, abrasive wear is also observed on the *A_γ_*, *A_α_*, and *A’_α_* surfaces. The tool’s wear is equal to *VB* = 0.5 mm. No built-up effect occurring could be the reason for the low temperature of the turning process and the low plasticity of the machined material.

In Figure 7a–d, cutting inserts after turning with *f* = 0.1 mm/rev of, respectively, the FEPA 046 sample and FEPA 100 sample are shown. The abrasive wear is observed on the *A_γ_*, *A_α_,* and *A’_α_* surfaces in both examples. There is no built-up effect on these cutting inserts. The crater is observed on the cutting edge of cutting inserts used during the turning of the FEPA 046 sample (Figure 7b). On the cutting edges of both cutting inserts, mechanical wear is observed. The value of the tool’s wear is equal to *VB* = 0.73 mm. The tool’s wear of the cutting insert used in the turning of the FEPA 100 shaft is equal to *VB* = 0.51 mm.

Similar signs of tool wear could be observed after turning with feed rate *f* = 0.125 mm/rev (Figure 8). The cutting insert after turning the FEPA 046 shaft is shown in Figure 8a,b. There is abrasive wear on the *A_α_* and *A’_α_* surfaces. The value of the tool’s wear is equal to *VB* = 0.76 mm. Except for abrasive wear, mechanical wear could be observed on the cutting edge of the cutting insert after turning the FEPA 100 sample (Figure 8c,d). The tool’s wear is equal to *VB* = 0.48 mm, which is the smallest value of that parameter from FEPA 100 tests.

Figure 9a,b show cutting inserts after the turning of the FEPA 046 sample with feed rate value *f* = 0.15 mm/rev. The abrasive wear could be observed. There is no built-up effect occurring. The abrasive and mechanical wear could be observed in Figure 9c,d, where the cutting insert after turning the FEPA 100 sample is shown. A crater on the cutting edge could also be noticed. There was a furrowing phenomenon observed (Figure 9d). The value of the tool’s wear of cutting inserts used during the turning of the FEPA 046 shaft is the smallest value for these parameters, and it is equal to *VB* = 0.7 mm, while the tool’s wear of the cutting insert used in the turning of the FEPA 100 shaft is equal to *VB* = 0.53 mm.

It could be observed that turning an MMC in conventional conditions has a significant influence on tool wear. The values of that parameter and signs of tool wear could be reasons for difficult machining conditions. Results of measurements of the value of tool wear are unsatisfying in terms of machining economy. These phenomena confirm the necessity to continue studies on the technology of machining MMCs for achieving satisfying results.

The observations of optical microscope measurements were compared with scanning electron microscope (SEM) TESCAN VEGA 5135 (TESCAN, Brno, Czech Republic) measurements of cutting inserts used for turning the FEPA 046 sample (Figure 10a,b) and FEPA 100 sample (Figure 10c,d). The cutting inserts used did not conduct electricity. It was necessary to carburize them. For that reason, chemical analysis of cutting insert composition could be less precise. Generally, the sticking of Al was observed on the *A_γ_*, *A_α_,* and *A’_α_* surfaces. Particles of ceramic could also be noticed. It could be observed that the tool’s wear after turning the FEPA 046 shaft is more significant than after turning the FEPA 100 shaft. The shaft made of the FEPA 046 composite is more hard-to-cut material than the FEPA 100 shaft. The influence of the grade of grains on the tool’s wear could be proved. Shabani et al. showed that hard Al_2_O_3_ reinforcing particles of the MMC material polished the surfaces of cutting inserts made of PCD [39]. Meanwhile, Durante et al. revealed the little influence of the size and hardness of abrasive particles on tool life [40]. Tool wear presented in this paper studies was manifested mostly by abrasive wear. The particles of ceramic reinforcement in FEPA 046 are larger than in FEPA 100. For that reason, the tool’s wear signs are more noticeable after turning the FEPA 046 shaft. It was observed that larger particles of ceramic reinforcement caused a mechanical impact on cutting inserts. There is no simple correlation between the tool’s wear *VB* and feed rate *f* that the machined material’s heterogeneity could cause.

Figure 11 presents the graph with the results of the tool’s wear of cutting inserts used during FEPA 046 (Figure 11a) and FEPA 100 turning (Figure 11b).

### 3.2. Geometrical Structure of Surface after Machining of Metal Matrix Composite

Three-dimensional surface roughness measurements of the workpiece were carried out for each sample without machining and after machining with four different values of feed rate. Sa and Sz parameters were measured because they are the most useful in the industry. Figure 12 shows the results of measurements of samples FEPA 046 (Figure 12a) and FEPA 100 (Figure 12b) before turning. In this case, the scale of 3D measurements is limited to 100 µm. It is a result of the necessity to compare samples with different sizes of grades. After turning, the 3D measurements of the samples’ surface roughness are shown on a scale limited to 50 µm for a more precise image. Before turning the FEPA 046 sample, the value of Sa reached 3.78 µm. The value of the Sz parameter before turning reached 17.04 µm. The value of Sa before turning the FEPA 100 sample reached 2.92 µm, and Sz was equal to 13.89 µm. The results of the Sa and Sz parameters are not satisfying because of the requirements which conform to machine parts used in the automotive and aeronautical industries.

The lowest value of Sa after turning shaft FEPA 046 (Figure 13a) was found after turning with *f* = 0.05 mm/rev, and it was equal to 1.23 µm. Similar to Sa, the smallest value of Sz was found after turning with *f* = 0.05 mm/rev, and it was equal to 6.46 µm. Figure 13b shows the results of turning the FEPA 100 sample with *f* = 0.05 mm/rev. The lowest value of Sa (Figure 13b) was equal to 0.92 µm. Similar to Sa, the lowest value of Sz was found after turning with *f* = 0.05 mm/rev, and it was equal to 5.40 µm. The periodic distribution of inequalities connected with the value of feed rate *f* is not observed. That phenomenon is not corresponding with the kinematic-geometric mapping of the machining process.

As was expected, increasing the feed rate f caused growth in machined surface roughness. Results of roughness measurements of the machined surface of the FEPA 046 shaft after turning with *f* = 0.1 mm/rev are shown in Figure 14a. Sa value reached 1.54 µm, and Sz value reached 7.84 µm. Turning of the FEPA 100 sample, compared with the FEPA 046 sample, is characterized by the lowest surface roughness values. The turning process of the FEPA 100 sample with feed rate *f* = 0.1 mm/rev (Figure 14b) caused the increase in the Sa parameter to 1.16 µm and the increase in Sz to 6.18 µm.

The turning process of the FEPA 046 sample with feed rate *f* = 0.125 mm/rev (Figure 15a) caused the increase in the Sa parameter to 2.08 µm and the increase in Sz to 10.54 µm. A further increase in feed rate *f* while turning the FEPA 100 Al_2_O_3_ ceramic shaft allows observation of change in the roughness distribution trend (Figure 15b). Sa value reached 1.32 µm, and Sz value reached 6.99 µm.

After the turning of both samples with feed rate *f* = 0.15 mm/rev, it could be observed that Sa and Sz reached the highest values. In the case of turning the FEPA 046 shaft with *f* = 0.15 mm/rev (Figure 16a), Sa and Sz reached, respectively, 2.60 µm and 12.56 µm. In Figure 16b, roughness measurements of the machined surface of the FEPA 100 shaft after turning with *f* = 0.15 mm/rev are shown. Sa and Sz values after turning with *f* = 0.15 mm/rev (Figure 16b) reached, respectively, 1.23 µm and 6.40 µm. A lack of kinematic-geometric projection of the cutting tool on machined material could result from the samples’ high graininess and grain extraction during the turning process.

Dyzia [41] noticed that the machined surface of MMCs was treated by picking and deforming. For that reason, the scratches and the characteristic accumulations were observed. Some particles of ceramic reinforcement were crushed, and some were pressed into the metal matrix. Contrary to the research presented in this article, no tearing off of SiC particles was observed. Generally, it could be observed that the Sa and Sz parameters increased after increasing the feed rate *f* (Figure 17a,b). This correlation is disturbed by the roughness measurements of the FEPA 100 sample after turning with *f* = 0.15 mm/rev. Values of Sa and Sz on this test section were lower than after turning with *f* = 0.125 mm/rev.

### 3.3. Forces in Machining Metal Matrix Composite

Figure 18 shows the values of the cutting force *F_c_* depending on the feed rate *f*. The values shown are the arithmetic mean of three tests carried out with the same parameters. In Figure 18a, it can be observed that the lowest value of cutting forces (*F_c_* = 11.79 N) of the machining of the FEPA 046 sample was noticed during the process with feed rate f = 0.1 mm/rev and *f* = 0.15 mm/rev. The highest value of that parameter (*F_c_* = 13.18 N) is observed during turning with *f* = 0.05 mm/rev. Figure 18b shows the results of the same test but for the turning of the FEPA 100 sample. The highest value of the cutting forces *F_c_* was noticed during turning with *f* = 0.05 mm/rev, and it is equal to *F_c_* = 17.95 N. The lowest value of the cutting forces of the machining of the FEPA 046 sample was noticed during the process with feed rate *f* = 0.1 (mm/rev), and it is equal to *F_c_* = 10.93 N. The decrease in the cutting force *F_c_* with the increase in the feed rate *f* is related to the increase in the susceptibility of the ceramic reinforcement grains to detachment from the material structure with the increase in feed rate *f* value.

During the turning of aluminum matrix composites, an increase in cutting forces was noticed (Figure 19). It is important to mention that in all cases, cutting force increased during the process. Intensive wedge wear was the cause of this effect. The highest value of cutting force *F_c_* was noticed near 10 s of the cutting process. There is a peak of value. It could be explained by the specific structure of the composite and the result of the presence of a large grain of the reinforcement phase. In comparison with Batista et al.’s [42] and Salguero et al.’s [43] studies, the obtained values of the cutting forces were small. Due to the constant cutting speed of *v_c_* = 35 [m/min] and the change in the shaft diameter with each subsequent pass, the cutting force *F_c_* values are higher with a lower feed rate *f*.

After analysis of the graph (Figure 20a), it could be noticed that the value of the Sa parameter increases with the increase in *VB* and with the decrease in *F_c_*. An inverse correlation is shown in Figure 20b. It could be stated that the Sz parameter increases with the decrease in *VB* and with the increase in *F_c_*. Machined material had a heterogeneous structure. Larger grains of ceramic reinforcement were connected weaker with the metal matrix than smaller grains. For that reason, a lower value of cutting force *F_c_* was needed to remove greater grains. However, due to the size of ceramic grains in this case, the Sa parameter of the surface roughness had a higher value. Tool wear was very high, given the short cutting time *s*. An increase in that parameter had a negative influence on surface roughness.

## 4. Conclusions

The following conclusions can be drawn from the results of this research:Structure of the metal matrix composite built of Al_2_O_3_ sinter saturated with an EN AC-44000 AC-AlSi_11_ alloy causes problems with the machining of this material.Value of the feed rate *f* influences the geometrical structure of the MMC after conventional turning.Tool’s wear of cutting inserts used during tests is substantial. It is not possible to describe some correlations between the feed rate value and tool wear value, but abrasive and mechanical wear including craters could be observed in most of the samples.The turning process contributed to the decrease in the Sa and Sz parameters of the machined surface compared to the not machined area. However, the results of this process are not satisfying. In general, some correlation between feed rate and surface roughness parameters could be observed. It can be stated that values of the Sa and Sz parameters increase with the growth of the feed rate’s value, except turning the FEPA 100 shaft with *f* = 0.15 (mm/rev). During this test, Sa and Sz values decreased compared to turning with *f* = 0.125 (mm/rev).Cutting force *F_c_* has low values. In this case, any correlations with feed rate *f* could not be evidenced.That correlation could not be noticed because of material heterogeneity. Reinforcement grains were pulled and crushed. The structure of the tested material caused the turning process to be uneven and results challenging to connect.The results of the studies described in this article show that the conventional method of turning MMCs is not satisfactory in terms of technological results. The tool’s wear was significant, and the surface roughness of the machined material was not satisfactory. The described tests were mainly focused on the results and possibilities of the conventional turning of MMCs.The research showed results of machinability in the conventional condition of MMC casting by the vacuum method with shaped reinforcement. Machining of that material was not described in the available literature.The carried out studies showed the necessity of continuing research on turning metal matrix composites built of Al_2_O_3_ sinter saturated with an EN AC-44000 AC-AlSi_11_ alloy. In future studies, LAM should be applied.

## Figures and Tables

**Figure 1 materials-15-08375-f001:**
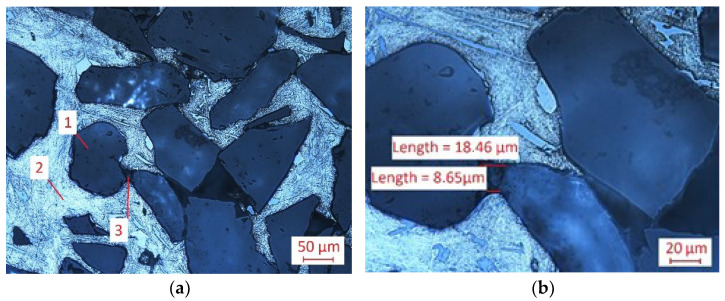
The metallographic examination of FEPA 100: (**a**) 1—Al_2_O_3_ sinter (reinforcement), 2—AlSi_11_ alloy (matrix phase), 3—porosity (50 µm), and (**b**) enlarged porosity (20 µm); and FEPA 046: (**c**) 1—Al_2_O_3_ sinter (reinforcement), 2—AlSi_11_ alloy (matrix phase), 3—porosity (100 µm), and (**d**) enlarged porosity (50 µm).

**Figure 2 materials-15-08375-f002:**
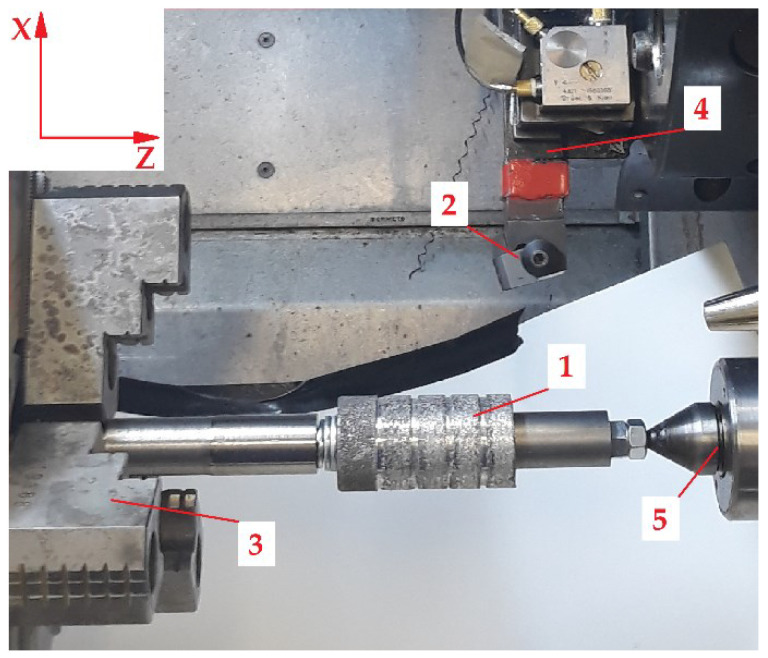
The MMC shaft in the lathe spindle (the coordinate system of the machine is shown): 1—sample shaft, 2—cutting insert, 3—chucks, 4—dynamometer, 5—tailstock.

**Figure 3 materials-15-08375-f003:**
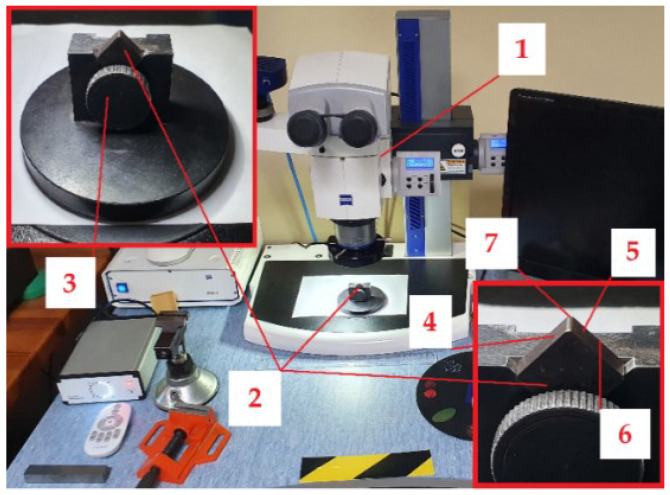
The tool’s wear measurements on a laboratory station with ZEISS SteREO Discovery.V20: 1—optical microscope, 2—cutting insert, 3—holder, 4—*A_α_* (principal flank surface), 5—*A′_α_* (auxiliary flank surface), 6—*Aγ* (rake surface—invisible), 7—cutting edge.

**Figure 4 materials-15-08375-f004:**
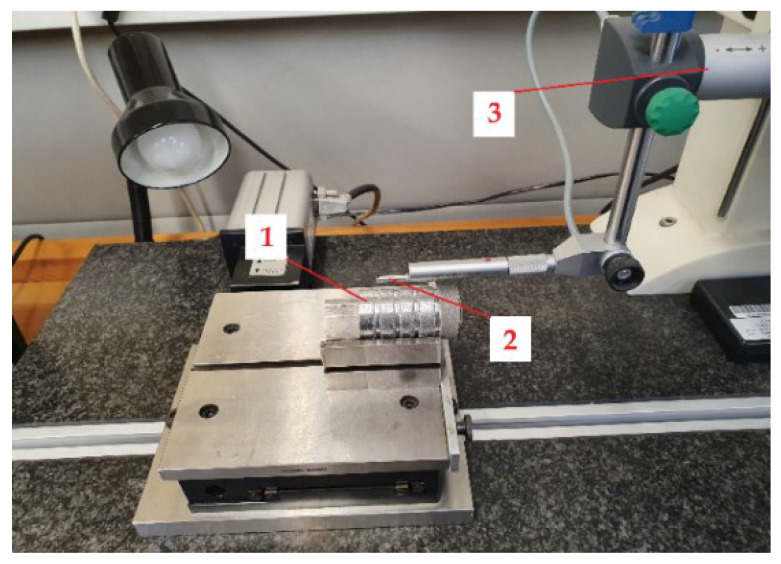
Three-dimensional surface roughness measurements with HOMMEL-ETAMIC T-8000: 1—sample shaft, 2—measurement head, 3—profilometer.

**Figure 5 materials-15-08375-f005:**
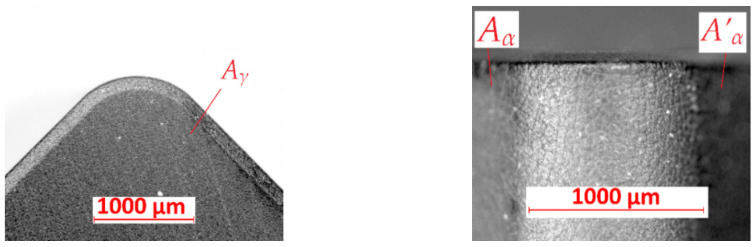
View of measurement areas for cutting inserts KENNAMETAL SNGN 120408 T01020 KYS25 before turning: *A_γ_*—rake surface, *A_α_*—principal flank surface, *A’_α_*—auxiliary flank surface.

**Figure 6 materials-15-08375-f006:**
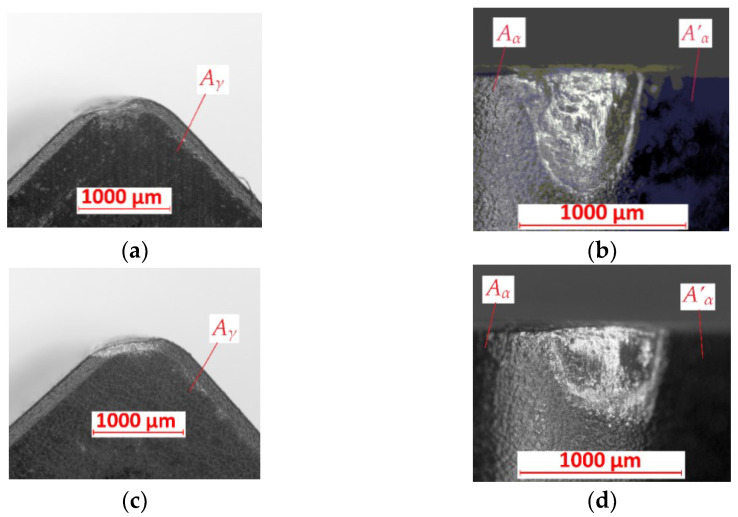
Results of tool’s wear measurements of cutting inserts after turning with *f* = 0.05 mm/rev of (**a**,**b**) FEPA 046 samples; (**c**,**d**) FEPA 100 samples: *A_γ_*—rake surface, *A_α_*—principal flank surface, *A’_α_*—auxiliary flank surface.

**Figure 7 materials-15-08375-f007:**
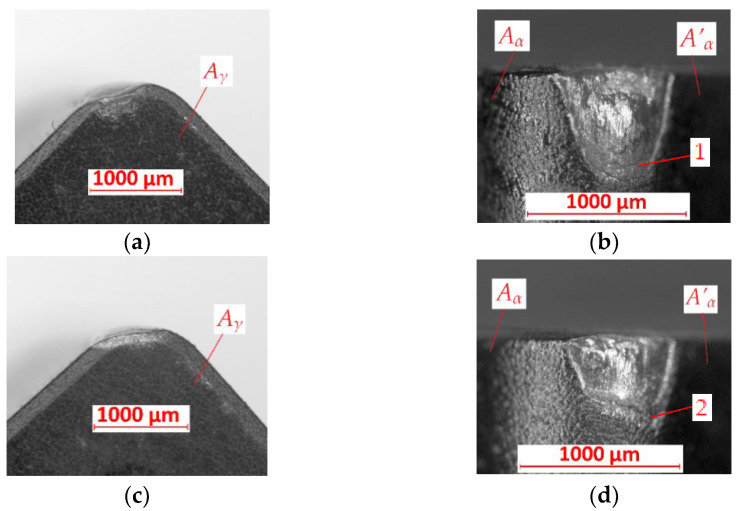
Results of tool’s wear measurements of cutting inserts after turning with *f* = 0.1 mm/rev of (**a**,**b**) FEPA 046 samples; (**c**,**d**) FEPA 100 samples: *A_γ_*—rake surface, *A_α_*—principal flank surface, *A’_α_*—auxiliary flank surface, *1*,*2*—places of abrasive wear.

**Figure 8 materials-15-08375-f008:**
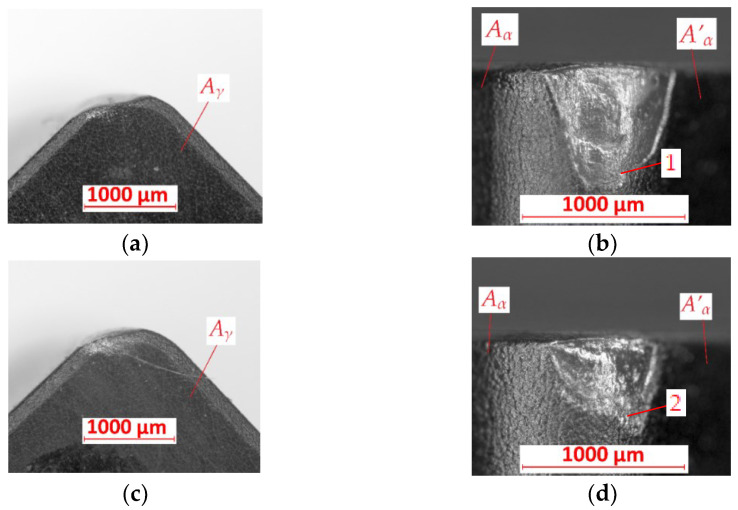
Results of tool’s wear measurements of cutting inserts after turning with *f* = 0.125 mm/rev of (**a**,**b**) FEPA 046 samples; (**c**,**d**) FEPA 100 samples: *A_γ_*—rake surface, *A_α_*—principal flank surface, *A’_α_*—auxiliary flank surface, *1*,*2*—places of abrasive wear.

**Figure 9 materials-15-08375-f009:**
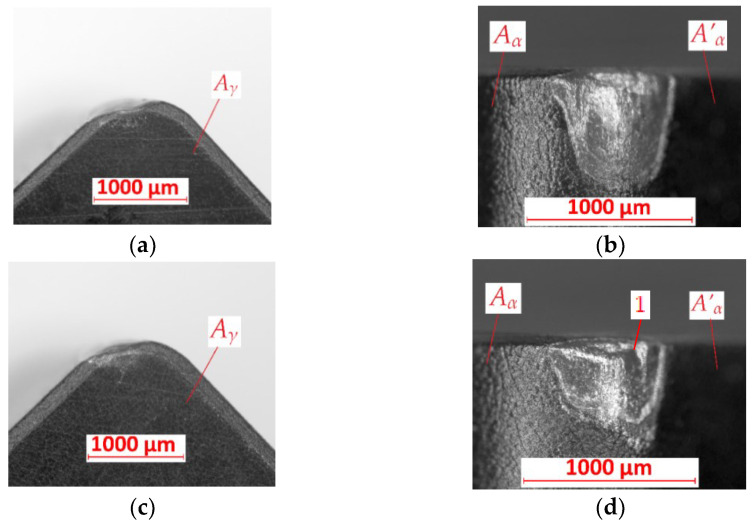
Results of tool’s wear measurements of cutting inserts after turning with *f* = 0.15 mm/rev of (**a**,**b**) FEPA 046 samples; (**c**,**d**) FEPA 100 samples: *A_γ_*—rake surface, *A_α_*—principal flank surface, *A’_α_*—auxiliary flank surface.

**Figure 10 materials-15-08375-f010:**
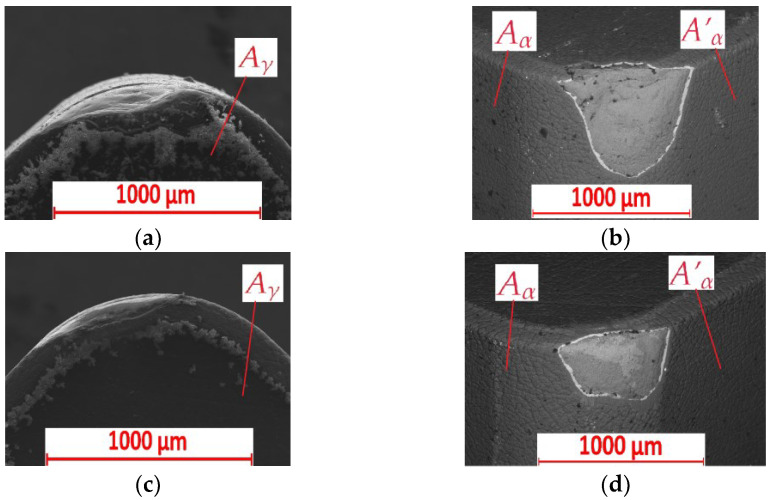
Results of SEM measurements of cutting inserts after turning of: (**a**,**b**) FEPA 046 samples; (**c**,**d**) FEPA 100 samples: *A_γ_*—rake surface, *A_α_*—principal flank surface, *A’_α_*—auxiliary flank surface.

**Figure 11 materials-15-08375-f011:**
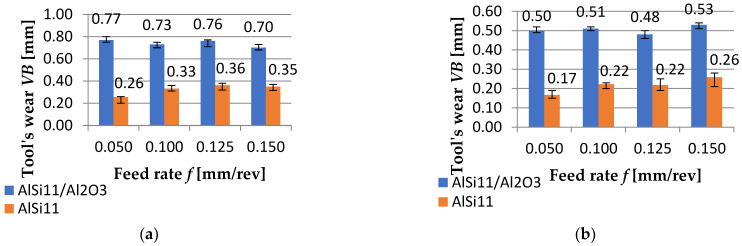
Results of measurements of tool’s wear: (**a**) for FEPA 046 and (**b**) FEPA 100 compared to AlSi_11_ (exact values are given above the bars).

**Figure 12 materials-15-08375-f012:**
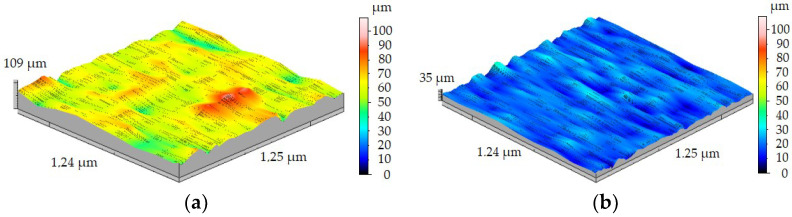
Results of 3D surface measurement samples before turning of: (**a**) FEPA 046, (**b**) FEPA 100.

**Figure 13 materials-15-08375-f013:**
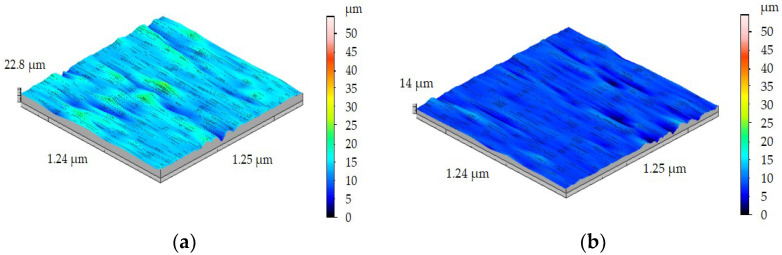
Results of 3D surface measurement samples after turning with *f* = 0.05 mm/rev of (**a**) FEPA 046, (**b**) FEPA 100.

**Figure 14 materials-15-08375-f014:**
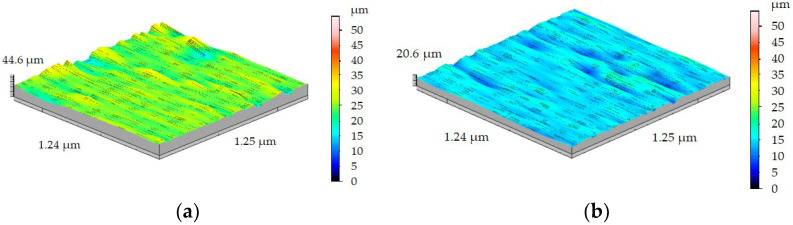
Results of 3D surface measurement samples after turning with *f* = 0.1 mm/rev of: (**a**) FEPA 046, (**b**) FEPA 100.

**Figure 15 materials-15-08375-f015:**
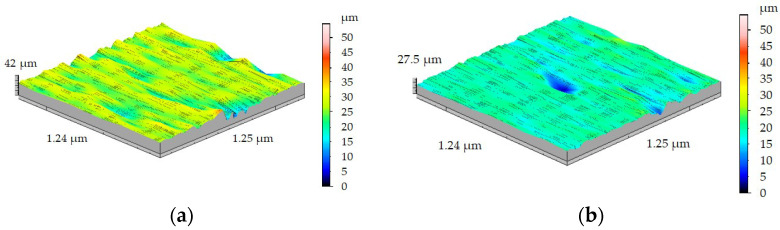
Results of 3D surface measurement samples after turning with *f* = 0.125 mm/rev of: (**a**) FEPA 046, (**b**) FEPA 100.

**Figure 16 materials-15-08375-f016:**
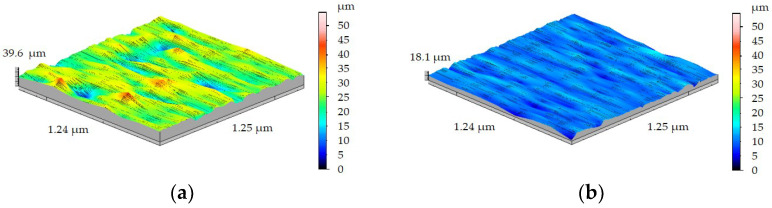
Results of 3D surface measurement samples after turning with *f* = 0.15 mm/rev of: (**a**) FEPA 046, (**b**) FEPA 100.

**Figure 17 materials-15-08375-f017:**
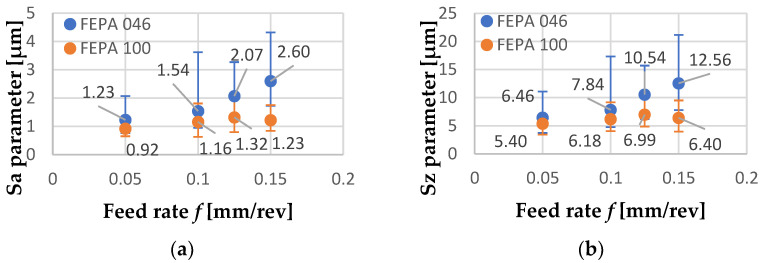
Results of machined surface roughness measurements (arithmetic average): (**a**) Sa parameter and (**b**) Sz parameter (exact values are given next to the bars).

**Figure 18 materials-15-08375-f018:**
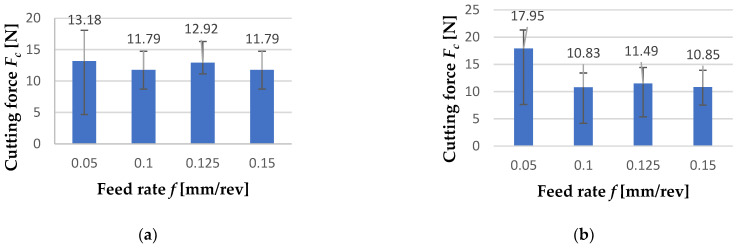
The graphs include results of forces measurements of turning of: (**a**) FEPA 046 shaft and (**b**) FEPA 100 shaft (exact values are given above the bars).

**Figure 19 materials-15-08375-f019:**
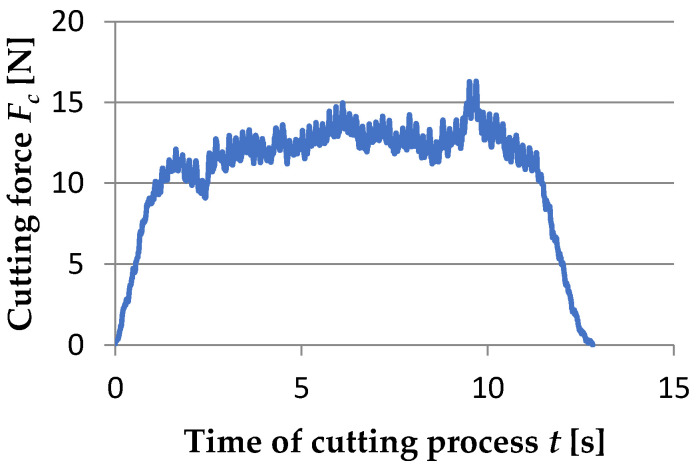
The graphs include exemplary results of forces in time of cutting process during turning of AlSi_11_/Al_2_O_3_ with *v_c_* = 35 m/min, *a_p_*= 0.15 mm, *f* = 0.125 mm/rev.

**Figure 20 materials-15-08375-f020:**
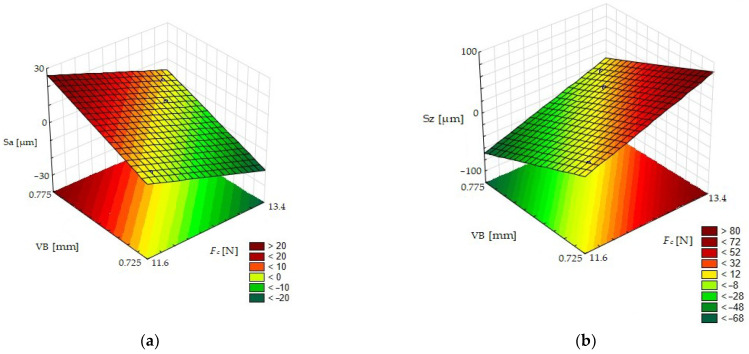
The surface roughness of MMC after turning as a function of tool’s wear *VB* and cutting force *F_c_*: (**a**) Sa parameter and (**b**) Sz parameter.

**Table 1 materials-15-08375-t001:** Geometry and properties of cutting insert Kennametal SNGN 120408 T01020 KYS25.

Insert shape	Insert Clearance Angle	Tolerance Class	Insert Features	Size	Thickness S	Nose Radius r_ε_	Chip Breaker	Coating
(-)	(°)	(mm)	(-)	(mm)	(mm)	(mm)	(-)	(-)
Square (S)	0 (N)	±0.013 (M)	G	12	4.76	0.8	T01020	KYS25
**Properties and application of insert material and coating**								
Mixed ceramic for hard machining, great hardness, thermal and wear resistance, excellent surface finish, lower cutting forces, and higher speeds; advanced TiCN CVD coating provides excellent chemical and depth-of-cut notch resistance.								

**Table 2 materials-15-08375-t002:** Technological parameters of the cutting process used for tests.

Type of Inserts	Sample	Symbol	*a_p_*	*v_c_*	*f*
(-)	(-)	(-)	(mm)	(m/min)	(mm/rev)
SNGN 120408 T01020 KYS25	FEPA 046	A1 A2 A3 A4	0.15 0.15 0.15 0.15	35 35 35 35	0.05 0.1 0.125 0.15
SNGN 120408 T01020 KYS25	FEPA 100	B1 B2 B3 B4	0.15 0.15 0.15 0.15	35 35 35 35	0.05 0.1 0.125 0.15

## Data Availability

Not applicable.

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
