# Peer review of "Tool Wear and Surface Roughness in Turning of Metal Matrix Composite Built of Al2O3 Sinter Saturated by Aluminum Alloy in Vacuum Condition"

_materials, 2022, doi:10.3390/ma15238375_

Round 1

Reviewer 1 Report

1. In Figures 5 to 9, the instruments and models used in the pictures are not described. Is it OM or SEM?

2. The description corresponding to Figures 7 and 8 refers to abrasive wear. Can you mark the parts where abrasive wear occurs? Or use large multiple observations to capture signs of abrasive wear.

 3. It is mentioned in the corresponding description in Figure 10 that the furrow phenomenon has been observed. Can you mark the parts where furrow wear occurs? Or the signs of furrow wear can be captured by using large multiple observations.

 4. Whether 3D surface measurements can optimize the reference plane? Since it is recommended to show the wave crest and wave trough in order to characterize the roughness. It is suggested that the values representing the height include positive and negative heights. For example, whether the reference plane can be optimized to - 20~30 μm for the height value of 0~50 μm in 3D surface measurements.

Author Response

The response in the attachment 

Reviewer 2 Report

The article submitted for review is of an important nature for the modern economy, related to the assessment of the wear of the cutting tool during the turning of composites. The authors have carried out an experimental assessment of the impact of turning composites on the wear rate of the cutting tool. The work, despite the high interest in this topic, has drawbacks.

1) The authors do not provide a mathematical description of their results. In this regard, it seems that the research was conducted for the sake of research. There is no mathematical model reflecting the relationship of wear with the type of material or forces formed in the cutting zone. The article should be supplemented with a description of the obtained or confirmed mathematical dependencies.

2) The absence of a formalized approach (mathematical models) in the article creates difficulties with the interpretation of the results obtained. So in Figure 18 (b), a graph is shown reflecting the results of the force measurement. From this graph it can be seen that as the feed increases, the force preventing cutting decreases. However, it is known from the metalworking technology that usually, with an increase in the feed per revolution, the area of the cut layer increases, and, as a result, the cutting force increases. The authors need to describe how they obtained (calculated) the averaged values of the forces presented in Figure 18, and also describe why these forces behave in this way.

3) The description of the results needs to be corrected, the authors contradict themselves in some places. two consecutive sentences refute each other 398 to 400, two consecutive sentences refute each other - "When turning composites with an aluminum matrix, the magnitude of the cutting forces was not observed (Fig. 19). It is important to note that in all cases the cutting force increases during the process."

4) In their conclusions, the authors claim the economic inexpediency of turning materials from composites, but they do not give any economic calculations. This point of conclusions can be removed.

Author Response

Response in the attachment
